# Semisynthetic Cardenolides Acting as Antiviral Inhibitors of Influenza A Virus Replication by Preventing Polymerase Complex Formation

**DOI:** 10.3390/molecules25204853

**Published:** 2020-10-21

**Authors:** Laurita Boff, André Schreiber, Aline da Rocha Matos, Juliana Del Sarto, Linda Brunotte, Jennifer Munkert, Flaviano Melo Ottoni, Gabriela Silva Ramos, Wolfgang Kreis, Fernão Castro Braga, Ricardo José Alves, Rodrigo Maia de Pádua, Cláudia Maria Oliveira Simões, Stephan Ludwig

**Affiliations:** 1Institute of Virology (IVM), Centre for Molecular Biology of Inflammation (ZMBE), Westfaelische Wilhelms University (WWU), 48149 Münster, Germany; laurita.boff@hotmail.com (L.B.); andre.schreiber@uni-muenster.de (A.S.); alinematos@hotmail.com (A.d.R.M.); julianaldelsarto@gmail.com (J.D.S.); brunotte@uni-muenster.de (L.B.); ludwigs@uni-muenster.de (S.L.); 2Laboratory of Applied Virology, Department of Pharmaceutical Sciences, Federal University of Santa Catarina (UFSC), Florianópolis, Santa Catarina 88040-900, Brazil; 3Respiratory Viruses and Measles Laboratory, Oswaldo Cruz Institute, Fiocruz, Rio de Janeiro 22775-051, Brazil; 4Department of Pharmaceutical Sciences, Faculty of Pharmacy, Universidade Federal de Minas Gerais, Belo Horizonte, Minas Gerais 31270-901, Brazil; fmottoni@hotmail.com (F.M.O.); gabsramos@outlook.com (G.S.R.); fernao.braga@gmail.com (F.C.B.); dylancover@gmail.com (R.J.A.); mpadua@hotmail.com (R.M.d.P.); 5Pharmaceutical Biology, Department of Biology, Friedrich-Alexander-University, 91054 Erlangen-Nuremberg, Germany; jennifer.munkert@fau.de (J.M.); wolfgang.kreis@fau.de (W.K.)

**Keywords:** semisynthetic cardenolides, anti-influenza, mechanism of action, polymerase activity inhibition

## Abstract

Influenza virus infections represent a major public health issue by causing annual epidemics and occasional pandemics that affect thousands of people worldwide. Vaccination is the main prophylaxis to prevent these epidemics/pandemics, although the effectiveness of licensed vaccines is rather limited due to the constant mutations of influenza virus antigenic characteristics. The available anti-influenza drugs are still restricted and there is an increasing viral resistance to these compounds, thus highlighting the need for research and development of new antiviral drugs. In this work, two semisynthetic derivatives of digitoxigenin, namely **C10** (3β-((*N*-(2-hydroxyethyl)aminoacetyl)amino-3-deoxydigitoxigenin) and **C11** (3β-(hydroxyacetyl)amino-3-deoxydigitoxigenin), showed anti-influenza A virus activity by affecting the expression of viral proteins at the early and late stages of replication cycle, and altering the transcription and synthesis of new viral proteins, thereby inhibiting the formation of new virions. Such antiviral action occurred due to the interference in the assembly of viral polymerase, resulting in an impaired polymerase activity and, therefore, reducing viral replication. Confirming the in vitro results, a clinically relevant ex vivo model of influenza virus infection of human tumor-free lung tissues corroborated the potential of these compounds, especially **C10,** to completely abrogate influenza A virus replication at the highest concentration tested (2.0 µM). Taken together, these promising results demonstrated that **C10** and **C11** can be considered as potential new anti-influenza drug candidates.

## 1. Introduction

Influenza viruses cause annual epidemics and occasional pandemics that affect thousands of people worldwide, thus representing a major public health concern [1]. Influenza viruses belong to the Orthomyxoviridae family, and based on their antigenic characteristics, they were subdivided into four genera: A, B, C, and D [2,3], but only A and B are clinically relevant to humans [4]. In addition, Influenza A viruses (IAVs) were further classified according to their surface glycoproteins: hemagglutinin (HA) and neuraminidase (NA) [5].

Influenza viruses contain an eight segmented negative-sense, single-stranded viral RNA genome that encodes for at least eleven essential viral proteins. Each RNA segment is encapsulated into a ribonucleoprotein (RNP) molecule, consisting of the nucleoprotein (NP) and a trimeric RNA-dependent RNA polymerase (RdRp) complex composed by PA, PB1, and PB2 [6]. Inside the nucleus, the RdRp carries out transcription and replication using the vRNA as a template for both transcription to mRNA and synthesis of complementary RNA (cRNA) that serves as the template for novel vRNAs. During the infection, the synthesis of mRNAs occurs prior to cRNA and vRNA transcription, which are then exported out of the nucleus for the synthesis of the viral proteins [7].

Although vaccination is the main prophylactic strategy used to decrease the impact of annual influenza epidemics [8,9], the effectiveness of licensed vaccines is rather limited due to the high mutation rate of the influenza virus genome that alters antigenic characteristics [10]. Thus, influenza vaccines need to be revised every year. Based on data gathered from north and south hemispheres, a scientific committee meets twice a year to define the influenza virus strains that will be included in the vaccine composition to be administered in the next flu season [11]. However, in case of a pandemic, vaccines have to be produced ad hoc and are generally available only some months after the emergence and spread of the new pandemic virus [12].

Additionally, anti-IAV drugs, such as M2 ion channel inhibitors (adamantane derivatives: rimantadine and amantadine, no longer recommended since 2004–2005 due to high resistance rates) and neuraminidase inhibitors (NAIs: oseltamivir and zanamivir), have been used worldwide. However, the emergence of influenza virus strains resistant to these drugs has been reported increasingly [13,14,15,16,17,18]. Because of the lack of effective antivirals, there are many new drug development programs all over the world. In Russia and China, a cell fusion inhibitor, named umifenovir, is clinically used [19,20]. Other NAIs (laninamivir and peramivir) were recently introduced in the pharmaceutical markets of Japan, China, South Korea, and USA for influenza prevention and control, but they do not seem to be more effective than oseltamivir. Thus, influenza treatment with these drugs cannot be considered satisfactory [10], and the development of new anti-influenza drugs with different mechanisms of action is a real need. Intending to reach this goal, recently two drugs that inhibit functions of the viral polymerase, baloxavir marboxil and favipiravir, have been licensed in specific markets (the first one in Japan and USA, and the second solely in Japan), but only for special groups of patients [21,22,23]. Based on the data available so far, these new drugs should be only considered for emergency use since baloxavir caused a high rate of resistance in clinical trials, whereas favipiravir was shown to be teratogenic [10]. Based on these facts, research and development (R&D) of new anti-influenza drugs remain valid and imperative.

Cardenolides are cardiac glycosides (CGs) mainly found in plant species of Apocynaceae (e.g., *Nerium oleander* L.) and Plantaginaceae (e.g., *Digitalis lanata* Ehrh. and *D. purpurea* L.) families [24], which have been broadly used for over 200 years to treat several cardiac conditions, including congestive heart failure, arrhythmias, and cardiogenic shock [25,26,27]. Therapeutically, the most relevant cardenolides are digoxin and digitoxin. Their mechanism of action and selectivity are mainly related to the direct inhibition of Na^+^/K^+^-ATPase α-subunit, which promotes cardiac muscle contraction [28].

In recent years, these compounds have been investigated for the treatment of other pathological conditions, such as viral infections. Their potential antiviral activity against a wide variety of viruses has been reported, such as adenovirus [29], chikungunya virus [30], coronavirus [31,32,33], cytomegalovirus [34,35,36], dengue virus [37], herpes virus [38,39,40,41], HIV [42,43,44], human papillomavirus [45], influenza virus [46,47,48], and respiratory syncytial virus [49].

Even though many studies have shown the potent antiviral activity of CGs, only three [46,47,48] focused on influenza viruses. Hoffmann et al. [46] reported that ouabain and lanatoside C reduced IAV and Influenza B virus (IBV) replication (strains A/WSN/33 (H1N1) and B/Yamagata/88, respectively) by regulating Na^+^ currents. Kiyohara et al. [47] showed that a CG isolated from the “desert rose” (*Adenium obesum*) reduced >50% viral titers of the IAV strain A/PR8/34 (H1N1). Likewise, Amarelle et al. [48] demonstrated that ouabain inhibited the protein translational machinery via a decrease of intracellular K^+^.

In this context, 16 new semisynthetic cardenolide derivatives of digitoxigenin (**C6a**, **C6b**, **C6c**, **C6d**, **C7a**, **C7b**, **C7c**, **C7d**, **C9**, **C10**, **C11**, **C12**, **C13**, **C14**, **C15**, and **C16**) were produced and preliminarily screened for anti-herpes activity as well as for cytotoxic effects against several human cancer cell lines [50]. Furthermore, the cytotoxic effects of some derivatives on H460 lung cancer cells were deeply investigated using different methodological approaches [51]. We also reported a proposal for the mechanism of anti-herpes action of **C10** and **C11**, which interfered with the intermediate and final steps of HSV replication, but not with the early stages since they completely abolished the expression of the UL42 (β) and gD (γ) proteins and partially reduced that of ICP27 (α); additionally, they were not virucidal and had no prophylactic effects [41].

In the present study, the anti-influenza virus effects of the same 16 new cardenolide derivatives were evaluated. Since two of them showed the most significant inhibitory potential on viral replication, namely **C10** (3β-((*N*-(2-hydroxyethyl)aminoacetyl)amino-3-deoxydigitoxigenin) and **C11** (3β-(hydroxyacetyl)amino-3-deoxydigitoxigenin), the main goal of this work was to explore their antiviral mechanism by using different in vitro and ex vivo methodological strategies. In this sense, **C10** and **C11** could be considered as promising candidates to be further evaluated for the treatment of influenza virus infections.

## 2. Results

### 2.1. Anti-Influenza Virus Activity of Cardenolide Derivatives

Firstly, 17 compounds (16 new semisynthetic cardenolide derivatives (**C6a** to **C16**) and digitoxigenin that was used as scaffold for the semisynthesis) were screened, at 1 µM, for their antiviral activity against IAV (strain A/WSN/33 (H1N1), multiplicity of infection (MOI) 0.01, 24 h) on A549 cells (Figure 1A). These compounds reduced viral titers at distinct levels, and those that reduced >50% of viral titers (digitoxigenin, **C7c**, **C10**, **C11**, and **C12**) were selected for further assays. A similar antiviral profile was observed when Madin-Darby canine kidney (MDCK) cells were used (Figure 1B). It is important to note that compounds **C10** and **C11** showed the highest reduction of influenza virus titers, when compared to the other compounds on both cell lines.

Furthermore, all compounds were screened against a broader panel of IAV and IBV strains (seasonal IAV H3N2 (Panama/2007/1999), H7N7 (Seal/Mass/1-SC35M/80), H1N1pdm (Hamburg/04/2009), and IBV (Lee/40)). They were also very active against all viral strains tested (Appendix A).

### 2.2. Cardenolide Derivatives Effects on Cell Viability and Influenza Replication

Since digitoxigenin and compounds **C7c**, **C10**, **C11**, and **C12** were shown to be the most active in the preliminary screenings, their cytotoxicity was evaluated on A549 and MDCK cells. None of them reduced cell viability at concentrations at least five times higher than the concentrations used in the initial screenings (Table 1). This evaluation ensured that the compounds would not act indirectly by simply reducing cell viability but rather act directly on virus propagation by interfering with certain stages of their replication cycle. For all compounds, IC_50_ and CC_50_ values as well as selectivity indices (SI) against IAV (strain A/WSN/33 (H1N1)) were calculated (Table 1). Compounds **C10** and **C11** were the most potent against IAV replication presenting IC_50_ values of approximately 0.06 µM on both cell lines. They showed SI values of 226 and 161 on A549 cells; and 306 and 241 on MDCK cells, respectively. Based on these favorable results, compounds **C10** and **C11**, whose chemical structures are depicted in Figure 2, were selected to further explore their antiviral mechanisms of action.

### 2.3. Anti-Influenza Virus Mechanism of Action of Compounds **C10** and **C11**

#### 2.3.1. **C10** and **C11** Affect the Expression of Viral Proteins at the Earlier Stages of Influenza A Virus Replication Cycle

The elucidation of the antiviral mechanism of action of **C10** and **C11** against IAV (A/WSN/33 (H1N1)) started by investigating the viral protein accumulation at different stages within the first replication cycle (Figure 3). Both compounds decreased early (NS1, NP, and PB1) and late viral protein expression (M1) when compared to the positive control (MDCK cells infected but not treated), at all time periods tested. Compound **C10** was most effective to reduce protein expression. In addition, impaired viral protein expression was already clearly observed at the earliest time point analyzed (2 h post-infection (p.i.)), demonstrating action of the compounds in very early stages of the replication cycle.

To analyze whether the interference of **C10** and **C11** with virus replication already occurs on the transcriptional level, the mRNA levels of influenza NP (segment 5) and M1 (segment 7) were quantified at 6 h p.i. Additionally, the amounts of newly produced cRNA and vRNA of the segments 5 and 7 were quantified (Figure 4). In agreement with the previous results, **C10** and **C11** decreased viral mRNA levels. Once again, **C10** presented the higher potency to reduce mRNA levels of both segments, when compared to the control group, especially for M1. As expected, a similar outcome was observed for cRNA and vRNA levels, with more pronounced differences after **C10** treatment.

Altogether, these results demonstrate that both compounds act at early stages of the IAV replication cycle by reducing the transcription of viral mRNA, leading to impaired synthesis of new viral proteins and, consequently, the production of new virions.

#### 2.3.2. **C10** and **C11** Affect the Polymerase Complex

To understand if the decrease of RNA levels was due to direct effects on viral polymerase, the activity of IAV polymerase after treatment with **C10** and **C11** was investigated using a plasmid-based mini genome polymerase assay. MDCK cells were transfected with plasmids expressing the constituents of the viral polymerase, PB1, PB2, PA, and NP, along with a plasmid that generates a viral RNA template expressing luciferase under control of the viral RNA promoters (Figure 5A). In this assay, the treatment diminished the polymerase activity as well. Noteworthy, the highest concentration of **C10** (1.0 µM) decreased the polymerase activity by >50%. While **C11** again was less potent and showed no significant reduction at the lowest concentration tested (0.5 µM), the compound still resulted in approximately 40% decrease of polymerase activity at 1.0 µM (Figure 5A).

Next, to explore if **C10** and **C11** were interfering with the assembly of the polymerase complex subunits, cells were infected with a recombinant IAV strain (Strep-PB2-WSN/H1N1) bearing a Strep-tag at the PB2 segment [52]. This allows specific precipitation of the polymerase complex containing vRNPs to analyze subunit composition.

The results demonstrated that both compounds, but more pronouncedly **C10**, interfered with the viral polymerase complex assembly, since PA, PB1 and NP amounts were strongly decreased in the complex precipitates (Figure 5B). Accordingly, it can be concluded that antiviral action occurred due to the interference in the assembly of viral polymerase complex resulting in a deficient polymerase activity and, therefore, reduced viral replication.

#### 2.3.3. **C10** and **C11** Reduce Viral Replication in an Ex Vivo Model of Human Lung Tissue

Finally, to evaluate antiviral activity of **C10** and **C11** in a more clinically relevant model of influenza virus infection, the compounds were tested in an ex vivo tumor-free human lung tissue explant model [53]. At 48 h p.i., **C10** reduced the viral titers by 4 logs at the lowest concentration tested (0.5 µM) and completely abrogated viral replication at the higher concentrations tested (1.0 and 2.0 µM) (Figure 6). At the same time, **C11** abolished viral replication only at the highest concentration tested (2.0 µM). Additionally, viability assay was performed with the treated lung tissues and confirmed that **C10** did not reduce tissue integrity up to 48 h p.i. In contrast, **C11** showed tissue toxicity at 48 h p.i. (>2-fold over control) (Appendix A).

## 3. Discussion

Cardenolides are molecules clinically used to treat heart diseases, such as heart failure, arrhythmias, and cardiogenic shock [25,26,27]. However, several studies have suggested promising effects for the treatment of other diseases [54], including those caused by viral infections [29,30,31,32,33,34,35,36,37,38,39,40,41,42,43,44,45,46,47,48,49]. Even though CGs have been investigated against many viruses, little is known about their potential anti-influenza activity. The available data support an anti-influenza activity in vitro [46,47,48] and in vivo using rodents that express a Na^+^/K^+^-ATPase α1 resistant to GC [48]. Therefore, the evaluation of new semisynthetic cardenolides is a valid strategy to identify new antiviral drugs against influenza virus infection.

The anti-influenza screening performed herein with the 16 new semisynthetic derivatives of digitoxigenin disclosed compounds **C10** and **C11** as the most potent ones with low IC_50_ values (ranging from 0.057 to 0.062 µM on A549 cells and 0.060 to 0.066 µM on MDCK cells, respectively), along with the highest SI values (ranging from 161 to 306, Table 1). As described by Boff et al. [50], digitoxigenin was used as scaffold for the semisynthesis of the derivatives cited above, and it was the most cytotoxic compound on A549 and MDCK cells, while presenting similar IC_50_ values to those of **C10** and **C11**. These results showed digitoxigenin as the tested compound with the lowest SI value, whilst **C10** and **C11** disclosed the highest SI values. A previous report showed that the cardenolide oleandrigenin-β-d-glucosyl (1→4)-β-d-digitalose, isolated from *Adenium obesum* (desert rose), presented anti-influenza activity since it reduced IAV titer (strain A/PR/8/34) on MDCK cells with an IC_50_ value of 0.86 µg/mL (=1.14 µM) [47]. It should be noted that this value is 19/17-fold higher than those here reported for **C10** and **C11**, respectively.

Our research group has recently described that both compounds inhibited Na^+^/K^+^-ATPase [50] and, therefore, this modulation function [55] might explain one of the mechanisms by which **C10** and **C11** interfere with influenza virus replication. The inhibition of Na^+^/K^+^-ATPase in host cells infected with influenza viruses caused by ouabain and lanatoside C has been already demonstrated by other research groups [46,56]. Recently, it was reported that ouabain decreased the intracellular K^+^ on A549 cells that is required for normal viral protein synthesis, and consequently reduced IAV replication due to Na^+^/K^+^-ATPase inhibition. Such a finding was confirmed in vivo by the inhibition of protein translation and the increased survival of influenza A/WSN/33-infected mice [48].

Both **C10** and **C11** demonstrated strong anti-influenza properties already manifested in the earlier stages of viral replication cycle. A significant reduction of viral proteins expression, such as PB1, NP, and NS1, was observed within only 2 h p.i. In accordance with this finding, **C10** and **C11** treatment also induced a decrease of viral mRNA levels, as they exemplarily demonstrated for segments 5 (NP) and 7 (M1). Furthermore, significant decreases in cRNA and vRNA levels were also detected. Altogether, these results suggest these compounds cause a general malfunction of the viral RNA-dependent RNA polymerase (RdRp), affecting its activity, which is involved in the synthesis of all three RNA species [57,58]. Indeed, the mini genome reporter gene assay employed herein indicated a decrease of the polymerase activity after treatment with both **C10** and **C11**, when compared to the positive control (Figure 5A).

The IAV polymerase complex is composed of PB1, PB2, and PA, that together with NP and vRNA, form the vRNP complex. For the proper function of the polymerase, a correct assembly of the polymerase subunits is required. In this study, it was demonstrated that both **C10** and **C11** interfered with the polymerase complex assembly since levels of the polymerase constituents (PA, PB1, and NP) decreased after the Strep-tag pulldown of PB2.

To continue studying **C10** and **C11**, the next step should be the investigation using an in vivo model to assess the promising antiviral effects already revealed in vitro. It is important to note that often in vivo studies are firstly done in non-human animals, such as mice or rats [59]. Nevertheless, the use of rodent models, even those xenotransplanted, is not advisable for this class of compounds, the cardenolides, since rodent cells are approximately 1000 times more resistant than human cells to the action of these compounds. Such difference can be explained because the main cellular target of cardenolides, the Na^+^/K^+^-ATPase enzyme, is expressed at different levels in humans and rodents [60,61]. Recently, Amarelle et al. [48] published a study using mice genetically modified to express Na^+^/K^+^-ATPase alpha 1 subunit, as occurs in humans, which can be an interesting approach to study cardenolides in vivo.

However, the use of an ex vivo model can be more robust and a viable economical alternative, simulating a bridge between in vitro and in vivo experiments and highlighting the non-use of laboratory animals. In this sense, an explant tumor-free human lung model provides a unique opportunity to analyze influenza viral replication in the complexity of human lung tissues that is as close to a real human infection scenario as possible. Since the lungs are the natural targets of these viruses, this model is extremely relevant for the investigation of potential new anti-influenza drugs.

Previously, it has been shown that an ex vivo infection is suitable for the evaluation of potential anti-influenza drugs and immunomodulatory agents. This model induced a complex innate immune response following the infection with the recombinant IAV (A/Panama/2007/1999 (H3N2)), which was manifested by a relevant expression of antiviral restriction factors, pro-inflammatory cytokines, interferons (IFNs) types I, II, and III, and a unique pattern of IFN-α subtypes. In this study, the macrolide antibiotic bafilomycin A (BafA) was applied to inhibit viral replication in lung tissues, and the pre-treatment with this drug for 1 h abolished IAV replication [53]. Complementarily, Weinheimer et al. [62] had already been shown that lung explants support the replication of diverse IAV strains.

Herein, the promising ex vivo results confirmed the in vitro findings demonstrating that when the human lung tissues were infected with the same recombinant IAV cited above, and treated with **C10** (1 µM) and **C10** and **C11** (2 µM), they reduced significantly the viral titers below the detection limit. Noteworthy, **C10** did not induce any relevant tissue toxicity, thus emphasizing its potential for further studies.

**C10** and **C11** also interfered with the assembly of viral polymerase resulting in a deficient polymerase activity. Such a mechanism can be the main antiviral action found herein for both compounds, and as far as we know, it has not been reported for any other CG. There are some recent studies reporting the inhibitory activity of influenza polymerase of different compounds, such as 5-(5-fluoro-1*H*-pyrrolo ((2,3-b)pyridin-3-yl)pyrazin-2(1*H*)-one derivatives [63], JL-5001 [64], R151785 [65], and for this reason, they are being called next-generation therapeutic agents.

## 4. Materials and Methods

### 4.1. Compounds, Cell Lines and Viruses

The semisynthesis and structure determination of new 16 cardenolide derivatives tested herein (**C6a**, **C6b**, **C6c**, **C6d**, **C7a**, **C7b**, **C7c**, **C7d**, **C9**, **C10**, **C11**, **C12**, **C13**, **C14**, **C15**, and **C16**) were carried out as previously described by Boff et al. [50]. Starting from digitoxigenin, 3β-azido-3-deoxydigitoxigenin was synthesized. Out of this intermediate compound, the two new series were prepared containing either glycosylated triazoles, obtained through click chemistry with propargyl glycosides (series i: **C6a**, **C6b**, **C6c**, **C6d**, **C7a**, **C7b**, **C7c**, and **C7d**), or compounds substituted in the alpha carbonyl position with different amines (series ii: **C9**, **C10**, **C11**, **C12**, **C13**, **C14**, **C15**, and **C16**). The structure of all new derivatives was verified by ^13^C-NMR and ^1^H-NMR [50].

Adenocarcinomic human alveolar basal epithelial cells (A549) and Madin-Darby canine kidney type II cells (MDCK-II) were cultivated in Dulbecco’s modified Eagle’s Medium (DMEM) and Eagle’s minimum essential medium (MEM) (Sigma-Aldrich, St. Louis, MO, USA), respectively, both supplemented with 10% fetal bovine serum (FBS) (Merck, Darmstadt, Germany) and 1% penicillin/streptomycin (P/S) at 37 °C and 5% CO_2_ in humidified conditions.

Regarding the viruses tested, A/Hamburg/04/2009 (H1N1pdm09) was kindly provided by German National Reference Centre for Influenza (Brunhilde Schweiger, Robert-Koch Institute, Berlin, Germany). Recombinant A/Seal/Mass/1-SC35M/80 (H7N7) (SC35M), A/WSN/33 (H1N1) (WSN) and B/Lee/40 were generated from plasmids by using the bidirectional pHW2000 reverse genetics system [66]. Recombinant IAV strain A/Panama/2007/1999 (H3N2) was generously provided by Thorsten Wolff (Robert-Koch Institute, Germany), and recombinant Strep-PB2-A/WSN/33 was kindly supplied by Martin Schwemmle (University of Freiburg, Germany). Viruses were propagated on A549 and MDCK cells for 72 h and viral titers were determined by standard plaque assay on MDCK cells [67,68].

### 4.2. Anti-Influenza Screening

The anti-influenza screening was performed as previous described by Schreiber et al. [69]. Briefly, 2.5 × 10^5^ A549 or MDCK cells per well (12-well plates) were seeded and infected (MOI 0.01) for 30 min at 37 °C. Viruses were diluted in infection phosphate-buffered saline (PBS) (PBS supplemented with 1% P/S, 0.2% (*v*/*v*) Bovine Serum Albumin (BSA) 35%, 0.01% MgCl_2_, 0.01% CaCl_2_). Following, cells were washed with PBS and treated with the compounds (digitoxigenin, **C6a**, **C6b**, **C6c**, **C6d**, **C7a**, **C7b**, **C7c**, **C7d**, **C9**, **C10**, **C11**, **C12**, **C13**, **C14**, **C15**, and **C16**) at 1 µM diluted in infection DMEM or MEM (DMEM for A549 cells and MEM for MDCK cells, both supplemented with 1% P/S, 0.2% (*v*/*v*) BSA 35%, 0.01% MgCl_2_, 0.01% CaCl_2_). Extemporaneously, 0.15 µg TPCK-treated trypsin were added, and supernatants were harvested after 24 h incubation. Confluent MDCK cell monolayers (6-well plates) were infected with 10-fold dilution series of virus supernatants for 30 min at 37 °C. Then, virus inoculum was replaced by plaque medium (14.2% (*v*/*v*) 10× MEM, 0.3% (*v*/*v*) NaHCO_3_ (7.5%), 0.014% (*v*/*v*) DEAE-dextran (1%), 1.4% (*v*/*v*) 100× P/S, 0.3% (*v*/*v*) BSA (35%), 0.01% (*v*/*v*) CaCl_2_ (1%), 0.01% (*v*/*v*) MgCl_2_ (1%), 0.9% (*v*/*v*) Agar (3%), and extemporaneously 0.15 µg TPCK-treated trypsin). Cells were then incubated for 72 h at 37 °C (IAV) or 33 °C (IBV), and viral plaques were counted. Only for the compounds that reduced >50% of viral titers (digitoxigenin, **C7c**, **C10**, **C11**, and **C12)**, the IC_50_ values were calculated (concentration of each compound that reduced viral replication by 50%), by nonlinear regression of concentration-response curves.

### 4.3. Cytotoxicity Evaluation

A549 and MDCK cells were seeded in 96-well plates and grown overnight. Cells were treated with different concentrations of digitoxigenin, **C7c**, **C10**, **C11**, and **C12** (5.0, 2.5, 1.25, 0.625, 0.313, 0.156, and 0.078 µM) or dimethyl sulfoxide (DMSO) (0.1%), diluted in 100 µL of DMEM or MEM, respectively, for 24 h at 37 °C and 5% CO_2_. Next, 15 µL of sterile water or Triton X-100 10% (CytoSelect^TM^ LDH Cytotoxicity Assay Kit) were added to each well in DMEM or MEM, for 10 min, at room temperature. Supernatants were collected and stored at −80 °C until used. Supernatants (90 µL) were transferred to clean 96-well plates and 10 µL of LDH Reagent (CytoSelect^TM^ LDH Cytotoxicity Assay Kit) were added to each sample and incubated for 20 min. Quantification was performed using a spectrophotometer at 405 nm. The concentration of each sample that reduced cell viability by 50% was estimated as CC_50_.

### 4.4. Anti-Influenza Viral Mechanism of Action

#### 4.4.1. Western Blot Analyses

To evaluate whether **C10** and **C11** interfere with viral or cellular protein expression, western blotting experiments were performed. In brief, cells were infected with IAV (A/WSN/33/H1N1) or were transfected with a recombinant IAV (Strep-PB2-WSN/H1N1) [52]. After that, cells were treated with each compound, washed with PBS, and lysed in radioimmunoprecipitation (RIPA) buffer (25 mM Tris-HCl (pH 8), 137 mM NaCl, 10% glycerol, 0.1% SDS, 0.5% sodium deoxycholate, 1% NP-40, and 2 mM EDTA (pH 8)). RIPA protein lysates were cleared by centrifugation (220*× g*, 4 °C, 10 min), mixed with 5× Laemmli buffer, separated by SDS-PAGE 10%, blotted onto nitrocellulose membranes, and blocked with TBS-T buffer (50 mM Tris-HCl (pH 7.5), 150 mM NaCl, and 0.2% Triton X-100 supplemented with 3% BSA (*w*/*v*)). After blocking, membranes were incubated overnight with polyclonal anti-influenza A (IAV) PB1 (GeneTex, EUA), polyclonal anti-IAV PA (GeneTex, EUA), polyclonal anti-IAV NP (GeneTex, EUA), polyclonal anti-IAV M1 (GeneTex, EUA), polyclonal anti-IAV NS1 (GeneTex, EUA), monoclonal anti-Tubulin (T6199, Sigma-Aldrich, EUA), and monoclonal anti-Strep Tag (GT661, Sigma-Aldrich, EUA), all diluted in blocking buffer. Respective secondary antibodies were diluted in TBS-Tween 0.5%, added to the membranes, and incubated for 45 min. The measuring of protein signals was performed with chemiluminescence by using the Li-Cor Odissey^®^ Fc Imaging System. Analyses were performed with Image Studio™ software (version 5.2).

#### 4.4.2. Quantitative Real-Time RT-PCR

Total cellular RNA was isolated by using RNeasy kit (Qiagen, Germany). cDNA was synthesized from 1 μg of total RNA by using RevertAid H Minus M-MLV Reverse Transcriptase (Fermentas, Germany). The following primers were used: human glyceraldehyde 3-phosphate dehydrogenase (GAPDH) forward (5′-GCAAATTCCATGGCACCGT-3′) and reverse (5′-GCCCCACTTGATTTGGAGG-3′); IAV nucleoprotein (NP) vRNA forward (5′-GGCCGTCATGGTGGCGAATGAATGGACGGAGAACAAGGATTGC-3′) and reverse (5′-CTCAATATGAGTGCAGACCGTGCT-3′); cRNA forward (5′-CGATCGTGCCCTCCTTTG-3′) and reverse (5′-GCTAGCTTCAGCTAGGCATCAGTAGAAACAAGGGTATTTTTCTTT-3′); mRNA forward (5′-CGATCGTGCCCTCCTTTG-3′) and reverse (5′-CCAGATCGTTCGAGTCGTTTTTTTTTTTTTTTTTCTTTAATTGTC-3′); IAV matrix protein 1 (M1) vRNA forward (5′-GGCCGTCATGGTGGCGAATTGCAGGGAAGAACACCGATC-3′) and reverse (5′-CGTGAACACAAATCCTAAAAT-3′); cRNA forward (5′-AGGGAAGAATATCGAAAGGAAC-3′) and reverse (5′-GCTAGCTTCAGCTAGGCATCAGTAGAAACAAGGTAGTTTTTTAC-3′); mRNA forward (5′-TCCTAGCTCCAGTGCTGGTC-3′) and reverse (5′-CCAGATCGTTCGAGTCGTTTTTTTTTTTTTCATTG-3′) [70]. qRT-PCR was performed by using QuantiTect SYBR Green PCR kit (Qiagen, Hilden, Germany) [71], and data were acquired with LightCycler 480 Instrument II (Roche, Gruyère, Switzerland). Gene expression was normalized to the endogenous housekeeping control gene GAPDH and analyzed by using 2−ΔΔCT method [72].

#### 4.4.3. Reporter Gene Assay

This assay followed the procedures described by Schräder et al. [73]. Briefly, MDCK cells grown overnight in 12-well plates were transfected with a transfection mixture containing plasmids encoding PB1, PB2, PA, and NP, a polymerase I (Pol I)-driven plasmid transcribing an influenza A virus-like RNA coding for the reporter protein firefly luciferase to monitor viral polymerase activity. The transfection mixture also contained a plasmid pHW72-Luci carrying a luciferase gene in negative orientation flanked by the viral promoter sequences, which served to normalize variation in transfection efficiency. Medium was changed 4 h post-transfection (p.t.), **C10** and **C11** were added at 0.5 and 1.0 µM and incubated. Cells were lysed 24 h p.t. and luciferase activity was measured using Dual-Glo Luciferase Assay System (Promega, Madison, WI, USA). PB2 was omitted from the transfection mixture as a negative control. Relative light units (RLU) were normalized to protein concentrations determined with a standard Bradford assay [74], and the relative polymerase activity was depicted as fold induction with respect to MOCK positive control.

#### 4.4.4. Virus Infection of Human Lung Tissue Explants

This assay was conducted as described by Matos et al. [53] and the ethical approval was given by the Ethical Council of the Deutsche Ärztekammer (AZ: 2016-265-f-S). Firstly, tumor-free human lung tissue explants were obtained from patients undergoing lung surgery at the University Hospital Muenster, Germany. All patients have given their written consent. Pieces of lung tissue were recovered in Roswell Park Memorial Institute (RPMI-1640) medium on the day of surgery, and stored at 4 °C. Immediately after, tissues were cut in pieces of approximately 100 mg, transferred to 12-well plates containing medium and incubated overnight at 37 °C with 5% CO_2_. Then, tissue fragments were infected with recombinant IAV (A/Panama/2007/1999 (H3N2)) in the presence of RPMI-1640 medium (supplemented with 2 mM l-Glutamine, 1% P/S and 0.1% BSA). After 60 min, tissues were transferred to new 12-well-plates with medium containing **C10** and **C11**, at 0.5, 1.0, and 2.0 µM, and were incubated for 48 h. Supernatants were collected at 1, 24, and 48 h post-infection (p.i.), and stored at −80 °C for viral titer determination using plaque standard assay on MDCK cells [67,68].

### 4.5. Statistical Analyses

The mean values ± standard deviations (SD) are representative of three or more independent experiments. For the determination of IC_50_ values, nonlinear regression of concentration-response curves was used. Statistical analyses were performed by ANOVA followed by post hoc tests as indicated. All analyses were performed by using GraphPad Prism 7.00 Software, La Jolla, CA, EUA.

## 5. Conclusions

The semisynthetic derivatives of digitoxigenin **C10** and **C11** showed anti-IAV action by affecting transcription of viral mRNA and thereby altering expression of viral proteins at early stages of the replication cycle, leading to impaired formation of new virions. Such antiviral action occurred due to the interference with the assembly of the viral polymerase complex, resulting in a deficient polymerase activity and, therefore, reducing viral replication. Attesting all these results obtained in vitro, a clinically relevant model of influenza infection by using ex vivo human tumor-free explant lung tissue confirmed the potential of these compounds to completely abrogate influenza A virus replication at the highest concentration tested (2 µM) at 48 h, mainly for **C10**. In conclusion, our findings suggest that by their promising and pioneer results, **C10** and **C11** can be considered as potential anti-influenza drug candidates.

## Figures and Tables

**Figure 1 molecules-25-04853-f001:**
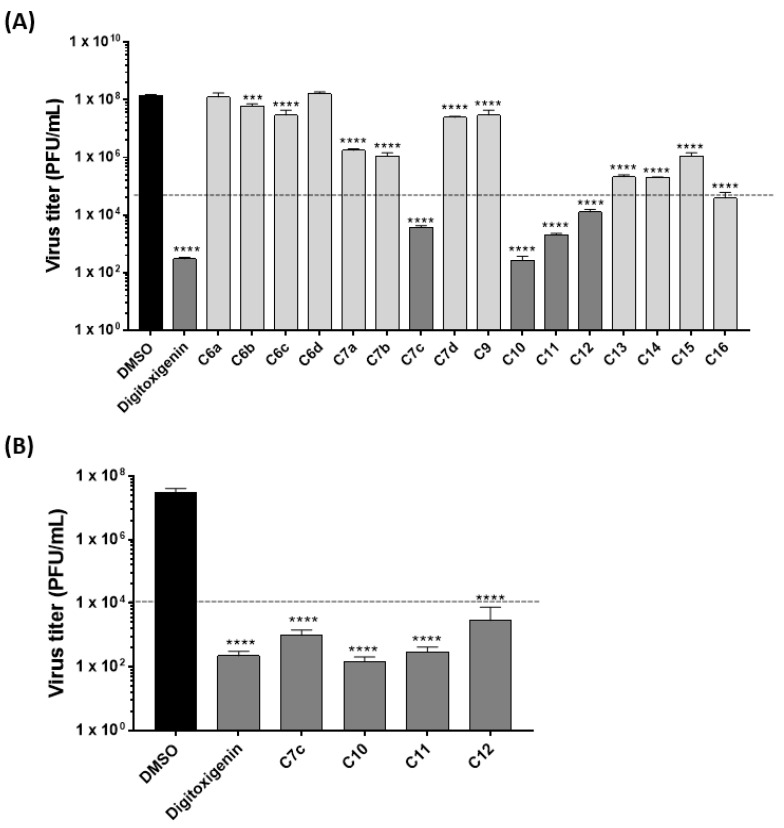
(**A**) Antiviral activity of new cardenolide derivatives (**C6a**, **C6b**, **C6c**, **C6d**, **C7a**, **C7b**, **C7c**, **C7d**, **C9**, **C10**, **C11**, **C12**, **C13**, **C14**, **C15**, and **C16**) and the scaffold digitoxigenin against influenza A virus (strain A/WSN/33 (H1N1)). A549 cells were infected at MOI 0.01 for 1 h and further treated with 1 µM of the compounds. Supernatants were collected at 24 h post-infection (p.i.) and titrated on Madin-Darby canine kidney (MDCK) cells. The black line indicates the compounds that reduced >50% of viral titer. Dimethyl sulfoxide (DMSO) served as solvent control and the titers of DMSO-treated cells were arbitrarily set to 100%. (**B**) Antiviral activity of the most active compounds in the preliminary screening (digitoxigenin, **C7c**, **C10**, **C11**, and **C12**) using the same experimental conditions cited above for (**A**) on MDCK cells. *** *p* < 0.001 and **** *p* < 0.0001 vs. DMSO, one-way ANOVA, post hoc test Dunnet.

**Figure 2 molecules-25-04853-f002:**
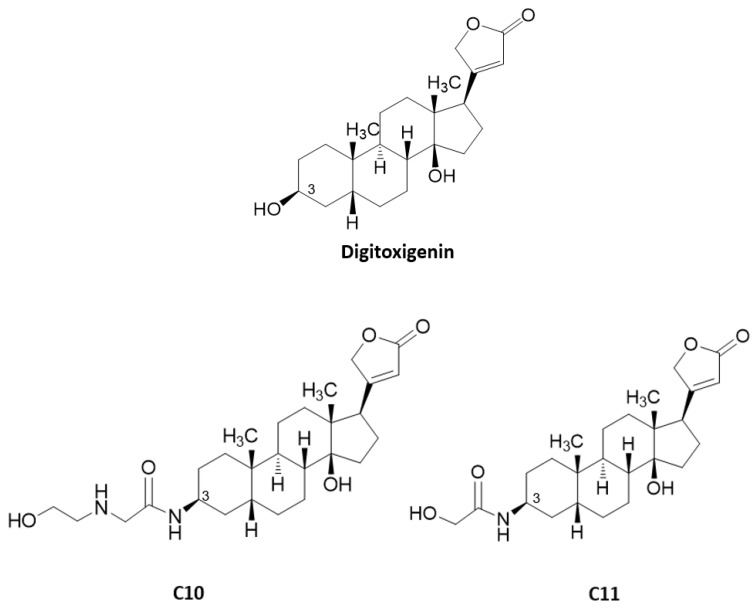
Chemical structures of digitoxigenin, **C10**, and **C11** derivatives.

**Figure 3 molecules-25-04853-f003:**
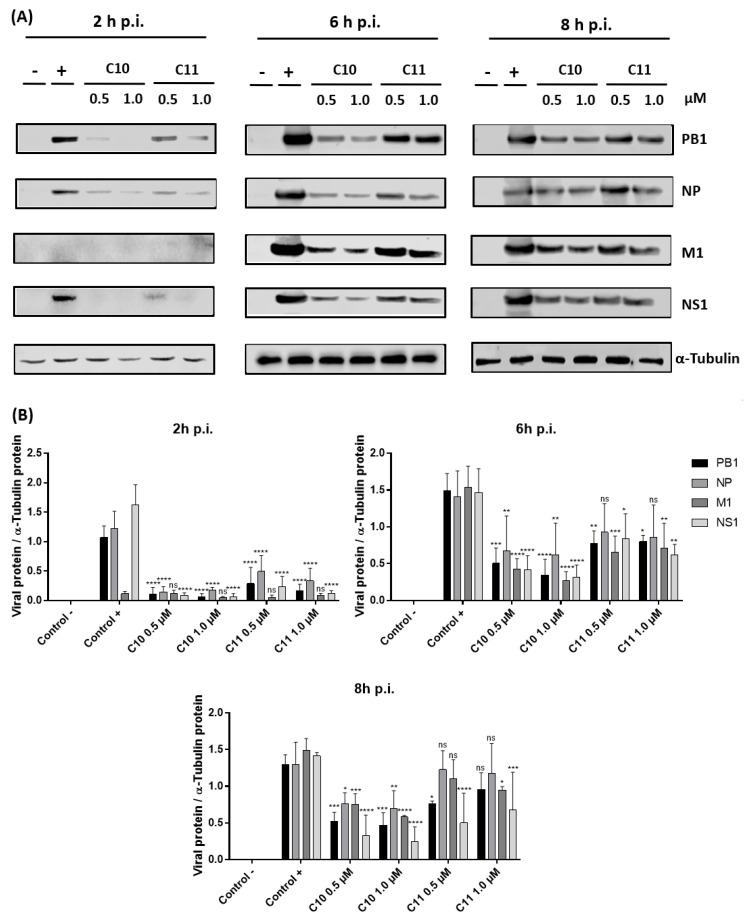
(**A**) Effects of **C10** and **C11** on influenza A virus (strain A/WSN/33 (H1N1)) protein expression over time. Western blot analysis occurred of MDCK cells infected at MOI 5 and then treated with both compounds (0.5 and 1.0 µM). Cell lysates were collected 2, 6, and 8 h post-infection (p.i.), run on SDS-PAGE 10%, and analyzed using specific antibodies for viral (PB1, nucleoprotein (NP), M1, and NS1) proteins. Equal protein loading was confirmed by probing for α-Tubulin. MDCK cells not infected or treated were used as negative control (−). MDCK cells infected and just DMSO-treated (solvent control) were used as positive control (+). (**B**) The graph indicates the ratio of each viral protein to α-Tubulin protein (ns = no significant, * *p* < 0.05; ** *p* < 0.01; *** *p* < 0.001; and **** *p* < 0.0001 vs. the respective viral controls, two-way ANOVA, post hoc test Dunnet).

**Figure 4 molecules-25-04853-f004:**
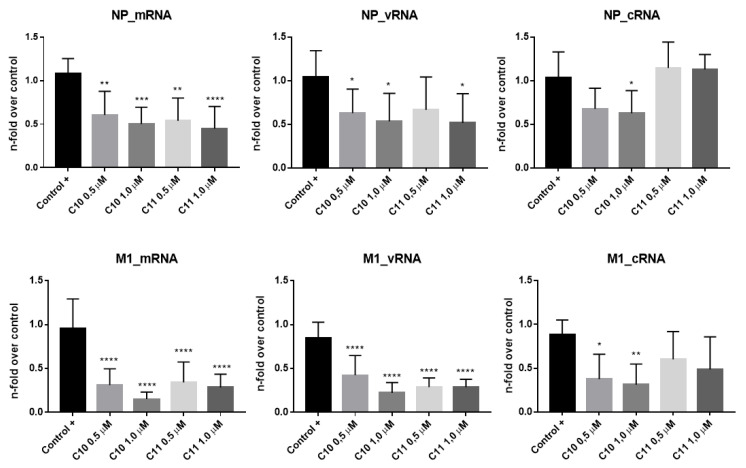
Effects of **C10** and **C11** on influenza A virus RNA transcription (strain A/WSN/33 (H1N1)). MDCK cells were infected at MOI 5 and then treated with both compounds (0.5 and 1.0 µM) for 6 h. Total RNA was extracted and viral mRNA, vRNA, and cRNA of NP (segment 5) and M1 (segment 7) were analyzed by qRT-PCR using glyceraldehyde 3-phosphate dehydrogenase (GAPDH) as the housekeeping gene. MDCK cells infected and just DMSO-treated (solvent control) were used as positive control (+). * *p* < 0.05; ** *p* < 0.01, *** *p* < 0.001 and **** *p* < 0.0001 vs. the respective viral controls, one-way ANOVA, post-hoc test Dunnet.

**Figure 5 molecules-25-04853-f005:**
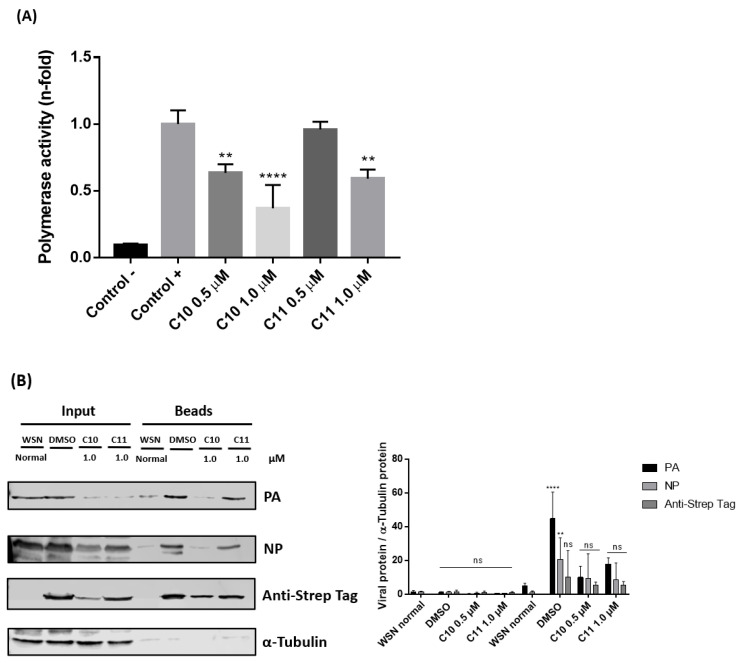
Effects of **C10** and **C11** on influenza A virus polymerase activity. (**A**) MDCK cells were transfected with plasmids expressing PB1, PB2, PA, and NP (PPPN), along with a plasmid-encoding luciferase under control (+) of viral promoter sequences. As negative control (−), the transfection mix lacking the PB2-encoding plasmid (PPN) was used. Medium was changed 4 h post-transfection (p.t.), **C10** and **C11** were added at 0.5 and 1.0 µM and incubated. After 24 h p.t., cells were lysed and the results were calculated as n-fold of luciferase activity in PPN-transfected **C10** and **C11**-treated cells, which was arbitrarily set to 1, and represent the mean ± SD of three independent experiments. ** *p* < 0.01 and **** *p* < 0.0001 vs. positive control, one-way ANOVA, post hoc test Dunnet. (**B**) Effects of **C10** and **C11** cardenolide derivatives on recombinant influenza A virus (Strep-PB2-WSN) protein expression, when bound or not to beads. Subcellular fractions were added to the beads and incubated overnight at 4 °C. After, beads were pelleted and Western blot analyses were performed using specific antibodies polymerase complex (PA, PB1, NP, Anti-Strep TagM1, and NS1) proteins. Equal protein loading was confirmed by probing for α-Tubulin. WSN normal = cells infected with influenza A virus ((strain A/WSN/33 (H1N1)) and not treated; DMSO = solvent control. The graph indicates the ratio of each viral protein to α-Tubulin protein (ns = no significant, ** *p* < 0.01, and **** *p* < 0.0001 vs. the respective viral controls, two-way ANOVA, post hoc test Dunnet).

**Figure 6 molecules-25-04853-f006:**
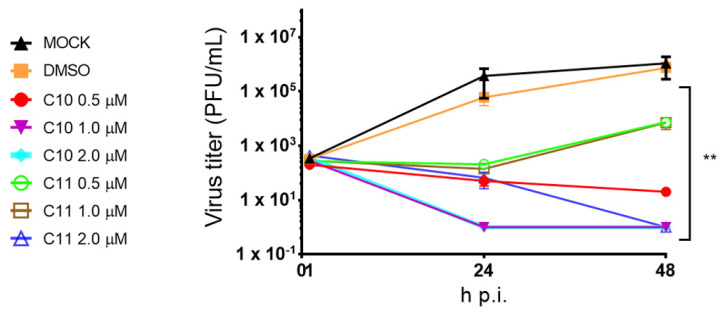
Effects of **C10** and **C11** on influenza A virus (A/Panama/2007/1999 (H3N2)) replication using an ex vivo human tumor-free lung model. Tissues were infected with 1 × 10^5^ plaque-forming units (PFU)/mL of this recombinant virus for 1 h and further treated with both compounds (0.5, 1.0, and 2.0 µM) for 48 h. Tissue supernatants were collected at 1, 24, and 48 h post-infection (p.i.) and titrated on MDCK cells. Each time point represents mean (±SD) of four independent experiments. ** *p* < 0.01 vs. MOCK, two-way ANOVA, post hoc test Dunnet.

**Table 1 molecules-25-04853-t001:** Cytotoxicity and antiviral activity of compounds **C7c**, **C10**, **C11**, **C12**, and digitoxigenin against influenza A virus (strain WSN/33 (H1N1)) replication on A549 and MDCK cells.

Compounds	A549 Cells	IAV (Strain WSN/33 (H1N1))		MDCK Cells	IAV (Strain WSN/33 (H1N1))	
	CC_50_ (µM)	IC_50_ (µM)	SI	CC_50_ (µM)	IC_50_ (µM)	SI
**C7c**	9.398 ± 0.741	0.125 ± 0.010	75.282	22.170 ± 2.393	0.103 ± 0.009	216.166
**C10**	12.790 ± 0.754	0.057 ± 0.002	226.052	18.343 ± 0.915	0.060 ± 0.004	306.164
**C11**	10.025 ± 0.962	0.062 ± 0.000	161.645	15.983 ± 0.235	0.066 ± 0.006	241.379
**C12**	13.280 ± 1.753	0.123 ± 0.013	107.647	23.340 ± 1.184	0.091 ± 0.008	257.483
**Digitoxigenin**	6.465 ± 1.004	0.070 ± 0.010	92.366	6.360 ± 0.427	0.072 ± 0.005	88.052

IAV: Influenza A Virus; CC_50_: 50% cytotoxic concentration (µM); IC_50_: 50% concentration that inhibited viral replication (µM). These values represent the mean ± SD of three independent experiments. SI: Selectivity index (SI = CC_50_/IC_50_).

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
