# Peer review of "Semisynthetic Cardenolides Acting as Antiviral Inhibitors of Influenza A Virus Replication by Preventing Polymerase Complex Formation"

_molecules, 2020, doi:10.3390/molecules25204853_

Round 1
Reviewer 1 Report
The manuscript investigated the ability of several semisynthetic derivatives of digitoxigenin (particularly, C10 and C11 derivatives) to inhibit influenza A virus infection in in vitro cell models as well as ex vivo human lung tissues. Both compounds reduced the expression of viral proteins at the early and late stages of replication cycle, and interfered with the assembly of viral polymerase resulting in an impaired polymerase activity, which impacted on virus replication.
The manuscript is well written, the used methods are appropriate, results were nicely presented, and discussion nicely addressed the obtained results.
comments
1- line 128: from .... not "form"
2- line 154: please write down the used concentrations.
3- Table 1 is missing.
4- line 215: were the lung tissues obtained specifically for this study or generally for other purposes? please indicate.
Author Response
Answers to the Reviewer 1 - Please see the attachment
_________________________________________
Summary: The manuscript investigated the ability of several semisynthetic derivatives of digitoxigenin (particularly, C10 and C11 derivatives) to inhibit influenza A virus infection in in vitro cell models as well as ex vivo human lung tissues. Both compounds reduced the expression of viral proteins at the early and late stages of replication cycle and interfered with the assembly of viral polymerase resulting in an impaired polymerase activity, which impacted on virus replication. The manuscript is well written, the used methods are appropriate, results were nicely presented, and discussion nicely addressed the obtained results.
COMMENT 1: line 128: from .... not "form"
Response 1: The correction was done now at line 340.
COMMENT 2: line 154: please write down the used concentrations.
Response 2: The suggestion was implemented in page11, lines 366 and 367.
COMMENT 3: Table 1 is missing.
Response 3: In the original manuscript submitted, Table 1 was there but when the journal production office prepared the manuscript to send to the reviewers it disappeared. We added Table 1 in the original manuscript (page 5).
COMMENT 4: line 215: were the lung tissues obtained specifically for this study or generally for other purposes? please indicate.
Response 4: The tumor-free human lung explants used in this study were obtained from patients undergoing lung surgery for different reasons at the University Hospital of Muenster. In addition to the specific purpose of each lung surgery, all patients gave their written consent to donate part of their lung tissues for scientific purpose. Ethical approval was given by the ethical council of the Deutsche Ärztekammer (AZ: 2016-265-f-S).

Reviewer 2 Report
The article of Laurita Boff et al. (Manuscript ID: molecules-963406) is interesting and well written and highlights the importance of studying natural compounds as a new source of antiviral drugs. I think that only few minor revisions are required.
MINOR Revisions:
1) In "Materials and Methods" section, how did the authors evaluate the protein concentration of their samples? Through Bradford assay?
2) Please provide in the text the Table 1 with cell viability values, IC50 and CC50.
3) Please provide the blot for PB1 protein in Figure 5B.
4) Which is the antiviral activity of Digitoxigenin, C7c, C10, C11 and C12 compounds against Influenza B virus (Lee/40) on A549 cells in Supplementary Figure 1?
Author Response
Answers to the Reviewer 2 - Please see the attachment
__________________________________________________________
Summary: The article of Laurita Boff et al. (Manuscript ID: molecules-963406) is interesting and well written and highlights the importance of studying natural compounds as a new source of antiviral drugs. I think that only few minor revisions are required.
COMMENT 1: In "Materials and Methods" section, how did the authors evaluate the protein concentration of their samples? Through Bradford assay?
Response 1: Yes, the protein concentrations were normalized by the standard Bradford assay (1976).
Bradford, M.M. A rapid and sensitive method for the quantitation of microgram quantities of protein utilizing the principle of protein-dye binding. Anal Biochem. 1976, 72, 248-254, https://doi.org/10.1016/0003-2697(76)90527-3.
COMMENT 2: Please provide in the text the Table 1 with cell viability values, IC50 and CC50.
Response 2: In the original manuscript submitted, Table 1 was there but when the journal production office prepared the manuscript to send to the reviewers it disappeared. We added the Table 1 in the original manuscript (page 5).
COMMENT 3: Please provide the blot for PB1 protein in Figure 5B.
Response 3: We appreciated the reviewer's suggestion, but as stated in the figure legend, the WSN normal lane represents the infection with a wt virus as a control for STREP specificity. In this sense, the DMSO lane must be compared to the treatment, and a clear reduction can be seen.
COMMENT 4: Which is the antiviral activity of Digitoxigenin, C7c, C10, C11 and C12 compounds against Influenza B virus (Lee/40) on A549 cells in Supplementary Figure 1?
Response 4: Multi replication cycle experiments of Influenza B virus (Lee/40) do not work on A549 cell line. For this reason, we just used MDCK cells.
Schreiber, A.; Liedmann, S.; Brunotte, L.; Anhlan, D.; Ehrhardt, C.; Ludwig, S. Type I interferon antagonistic properties of Influenza B virus polymerase proteins. Cell Microbiol. 2019, 11, e13143, https://doi.org/10.1111/cmi.13143.

Reviewer 3 Report
The manuscript ”Semisynthetic cardenolides acting as antiviral inhibitors of influenza A virus replication by preventing polymerase complex formation” concerns an urgent need of an effective therapy of influenza virus infection, which still remains elusive.
This is a very interesting work, well written with a broad spectrum of research methods. The main advantage of the study is the identification of the novel compounds, which may be considereted as a potential new anti-influenza drug candidates.
The structure of the manuscript is very good.
However, there are only several points that may be proven:
- The effects of C10 and C11 on viral plaque size may be added (microscopic pictures)
- C10 and C11 exhibits antiviral activities against all examined strains of IAVs as weel as previous work of Authors revealed anti-herpes activity of these coumpounds. It should be clarified if C10 and C11 are candidates for universal drug for various viral infection (multiple subtypes) or rather act selectivelly through precise molecular way?
This paper will be suitable for pubication in Molecules.
Author Response
Answers to the Reviewer 3 - Please see the attachment
__________________________________________________________
Summary: The manuscript “Semisynthetic cardenolides acting as antiviral inhibitors of influenza A virus replication by preventing polymerase complex formation” concerns an urgent need of an effective therapy of influenza virus infection, which still remains elusive. This is a very interesting work, well written with a broad spectrum of research methods. The main advantage of the study is the identification of the novel compounds, which may be considered as a potential new anti-influenza drug candidates. The structure of the manuscript is very good. However, there are only several points that may be proven:
COMMENT 1: The effects of C10 and C11 on viral plaque size may be added (microscopic pictures)
Response 1: We appreciated the reviewer's suggestion, but as the staining dye in this assay, we used the neutral red that was overlaid onto the agarose layer, which enabled the lysis plaques to be viewed and counted around after 1-3 h. After this time staining, these lysis plaques can no longer be viewed.
González-Hernández, M.B., Perry, J.W., Wobus, C.E. Neutral Red assay for murine norovirus replication and detection in a mouse. Bio Protoc. 2016, 3, e415.
COMMENT 2: C10 and C11 exhibits antiviral activities against all examined strains of IAVs as well as previous work of authors revealed anti-herpes activity of these compounds. It should be clarified if C10 and C11 are candidates for universal drugs for various viral infection (multiple subtypes) or rather act selectively through precise molecular way?
Response 2: Our research group has recently described that C10 and C11 inhibited Na+/K+-ATPase (Boff et al., 2019) and, therefore, this modulation function (Amarelle; Lecuona, 2018) might explain one of the mechanisms by which both compounds interfere with influenza virus replication. They also interfered with the assembly of viral polymerase resulting in a deficient polymerase activity. Such mechanism can be the main antiviral action found herein for these compounds, and as far as we know, it has not been reported for any other cardenolide.
The elucidation of the mechanism of antiherpes action of C10 and C11 showed that they interfere mainly with the late steps of HSV replication, i.e., viral replication, assembling, release of new viruses and viral intercellular propagation. In a minor extension, they also interfere with the intermediate stages of viral replication but not with the early steps. Additionally, they were not virucidal and had no prophylactic effects. (Boff et al., 2020),
Although C10 and C11 were able to inhibit Na+/K+-ATPase, depending on the virus tested, the steps of viral replication cycle affected will be different.
Boff L, Munkert J, Ottoni FM, Schneider NFZ, Ramos GS, Kreis W, et al. Potential anti-herpes and cytotoxic action of novel semisynthetic digitoxigenin-derivatives. Eur J Med Chem 2019; 167:546-561. http://dx.doi.org/10.1016/j.ejmech.2019.01.076.
Amarelle, L.; Lecuona, E. The antiviral effects of Na,K-ATPase inhibition: A minireview. Int J Mol Sci. 2018, 19, 2154, https://doi.org/10.3390/ijms19082154.
Boff, L., Schneider, N.F.Z., Munkert, J., Ottoni, F.M., Ramos, G.S., Kreis, W., Braga, F.C., Alves, R.J., Pádua, R.M., Simões, C.M.O. Elucidation of the mechanism of anti-herpes action of two novel semisynthetic cardenolide derivatives. Arch Virol. 2020, 165, 1385-1396, https://doi.org/10.1007/s00705-020-04562-1.
COMMENT 3: This paper will be suitable for publication in Molecules.
Response 3: Thanks!

This manuscript is a resubmission of an earlier submission. The following is a list of the peer review reports and author responses from that submission.
Round 1
Reviewer 1 Report
The manuscript by Boff et al. describes the characterisation of a series of semisynthetic cardenolides against Influenza A virus (IAV). There is an ongoing need for the development of anti-Influenza drugs and this series of cardenolide compounds have previously been described by the same group against herpes virus (Boff et al, Arch Virol. 2020; Boff et al., Eur J Med Chem, 2019).
Within the methods the cytotoxicity evaluation was performed on a remarkably low concentration curve only reaching 5 uM whilst in the results CC50’s are reported above these concentrations, up to 23 uM. These CC50’s are either incorrect or the concentrations are incorrectly reported which I cannot determine given the CC50 curves are not presented, in fact no concentration curves are presented within the manuscript.
The authors conclude that compounds C10 and C11 affect the expression of viral proteins at the early stages of the replication cycle. Whilst they did reduce the expression of proteins I am unable to interpret which proteins due to a lack of labels on Fig 3. Furthermore, the use of an infected + DMSO control would be appropriate to ensure there is no effect of the DMSO on protein expression.
Given this reduction in protein it is not surprising that there was a reduction in the mRNA present. Again the presence of a DMSO control would be appropriate.
A reporter gene assay was utilized to demonstrate that compounds C10 and C11 were acting directly on the polymerase complex. Following a pull-down and western blot (Fig. 5B) to investigate the subunit compositions the authors claim to observe a reduction in the polymerase complex however the western blot does not appear to show this. In fact, when compared with WSN (normal Strep-PB2-WSN infection) both PA and NP appear to increase significantly.
Labels on Fig 4 need attention.
Author Response
Answers to the Reviewer 1 - "Please see the attachment"
_________________________________________________________
Summary: The manuscript by Boff et al. describes the characterization of a series of semisynthetic cardenolides against Influenza A virus (IAV). There is an ongoing need for the development of anti-Influenza drugs and this series of cardenolide compounds have previously been described by the same group against herpes virus (Boff et al, Arch Virol. 2020; Boff et al., Eur J Med Chem, 2019).
COMMENT 1: Within the methods the cytotoxicity evaluation was performed on a remarkably low concentration curve only reaching 5 uM whilst in the results CC50’s are reported above these concentrations, up to 23 uM. These CC50’s are either incorrect or the concentrations are incorrectly reported which I cannot determine given the CC50 curves are not presented, in fact no concentration curves are presented within the manuscript.
Response 1: Due to the limitation regarding the available amount of the compounds, no concentration curves above to 23 uM were showed. In this sense, an estimation of the CC50 values, using a mathematical calculation, was done since none of them showed cytotoxicity at the highest concentration tested (5 µM). Such estimation was made based on eight tested concentrations (5.0, 2.5, 1.25, 0.625, 0.313, 0.156 and 0.078 µM) of digitoxigenin, C7c, C10, C11 and C12.
The text was modified to make it clearer and more understandable (in yellow the text added now):
Page 4, lines 159-160: “The concentration of each sample that reduced cell viability by 50% was estimated as CC50”.
COMMENT 2: The authors conclude that compounds C10 and C11 affect the expression of viral proteins at the early stages of the replication cycle. Whilst they did reduce the expression of proteins, I am unable to interpret which proteins due to a lack of labels on Fig 3. Furthermore, the use of an infected + DMSO control would be appropriate to ensure there is no effect of the DMSO on protein expression.
Response 2: In the original manuscript submitted, the labels were there but when the journal production office prepared the manuscript to send to the reviewers they disappeared. We corrected it in Figure 3 (page 8), and improved its legend to better explain the use of DMSO as follows (in yellow the text added now):
Figure 3. A) Effects of C10 and C11 on influenza A virus [strain A/WSN/33 (H1N1)] protein expression over time. Western blot analysis of MDCK cells infected at MOI 5 and then treated with both compounds (0.5 and 1.0 µM). Cell lysates were collected 2, 6 and 8 h p.i., run on SDS-PAGE 10% and analyzed using specific antibodies for viral (PB1, NP, M1 and NS1) proteins. Equal protein loading was confirmed by probing for α-Tubulin. MDCK cells not infected or treated were used as negative control (-). MDCK cells infected and just DMSO-treated (solvent control) were used as positive control (+). B) The graphs indicate the ratio of each viral protein to α-Tubulin protein (ns = no significant, * p < 0.05; ** p < 0.01; *** p < 0.001 and **** p<0.0001 vs the respective viral controls, two-way ANOVA, post-hoc test Dunnet).
COMMENT 3: Given this reduction in protein it is not surprising that there was a reduction in the mRNA present. Again, the presence of a DMSO control would be appropriate.
Response 3: We do feel that these experiments are important to show, since there would have been the possibility that the compound could act on protein translation from mRNA. Thus, showing differences in mRNA amounts adds important information towards the mode of action. Similar to Figure 3, we rearranged the legend of Figure 4 to improved it as follows (in yellow the text added now):
Figure 4. Effects of C10 and C11 on influenza A virus RNA transcription [strain A/WSN/33 (H1N1)]. MDCK cells were infected at MOI 5 and then treated with both compounds (0.5 and 1.0 µM) for 6 h. Total RNA was extracted and viral mRNA, vRNA and cRNA of NP (segment 5) and M1 (segment 7) were analyzed by qRT-PCR using GAPDH as the housekeeping gene. MDCK cells infected and just DMSO-treated (solvent control) were used as positive control (+). * p < 0.05; ** p < 0.01 and *** p < 0.001 vs the respective viral controls, one-way ANOVA, post-hoc test Dunnet.
COMMENT 4: A reporter gene assay was utilized to demonstrate that compounds C10 and C11 were acting directly on the polymerase complex. Following a pull-down and western blot (Fig. 5B) to investigate the subunit compositions the authors claim to observe a reduction in the polymerase complex however the western blot does not appear to show this. In fact, when compared with WSN (normal Strep-PB2-WSN infection) both PA and NP appear to increase significantly.
Response 4: There seems to be a misunderstanding here. As stated in the figure legend, the WSN normal lane represents the infection with a wt virus as a control for STREP specificity. Thus, the DMSO lane must be compared to the treatment, and a clear reduction can be seen. Accidentally, we kept the PB1 panel, although the WSN wt control indicated there seems to be an unspecific binding. This was now omitted, and the STREP blot was replaced by one with better quality.
COMMENT 5: Labels on Fig 4 need attention.
Response 5: As answered above in COMMENT 3, the legend of Figure 4 was modified to make it clearer and more understandable.

Reviewer 2 Report
The present study is focused on the investigation of the antiviral properties of novel semisyntetic cardenolides against different influenza strains. Among the tested compounds, C10 and C11 were highlighted to be the most promising compounds, exhibiting an anti-influenza power similar to that of the standard cardenolide digitoxigenin.
The study is of interest in the field and is well designed and developed. Particularly, details about the mechanisms involved in the inhibition of viral replication have been demonstrated.
The following points should be checked and clarified in order to improve the manuscript for publication.
- Introduction (page 3, lines 108-114): results should be not included in this section; conversely, the aim of the study and the impact of the expected results should be reported.
- Materials and methods (page 3, line 120): specify that A549 is an adenocarcinoma cell line.
- Materials and methods (page 4, line 140): the choice of the tested concentration for the antiviral activity should be explained.
- Materials and methods (page 4, line 150) “Only for the compounds that 148 reduced > 50% of viral titers (digitoxigenin, C7c, C10, C11 and C12), the IC50 values were calculated”: IC50 calculation requires that a concentration-response curve be made. Please, explain the method used for the calculation of this parameter.
- Materials and methods (page 4, line 163): change “Western” as western
- Materials and methods (page 5, line 214): specify the source of lung tissue explants and provide the ethical statement.
- Materials and methods (page 5, line 223): specify the method applied for IC50 calculation and software used. Moreover, the Authors should explain if data from different biological replicates are pooled for the statistical analysis and how many replicates have been made.
- Figure 1 (page 6): the statistical significance with respect the control should be included; moreover, that of C7c, C10, C11 and C12 with respect to the standard cardenolide digitoxigenin, should be included in Figure 1B. Accordingly, the statistical significance should be included in Figure S1.
- Digitoxigenin is usually effective like C10 against all the tested strains. Are there some advantages in using the synthetic derivative C10 with respect digitoxigenin? Please, discuss.
- 2. Cardenolide derivatives effects on cell viability and influenza replication (page 6): the CC50, IC50 and selectivity indices (SI) parameters should be described in the paragraph.
- Table 1 (page 7): the IC50 difference should be calculated. Are there significant differences between the IC50 of C10, C11 and digitoxigenin?
- Figure 4 (page 9): time exposure for studying the effect of the test compound on influenza A virus RNA transcription is lower than that applied for the viral replication (24 h). Please, explain.
- 3.2. C10 and C11 affect the polymerase complex (page 9): a comparison with digitoxigenin should be included
- Figures 3 and 5: densitometric analysis should be included
- ex vivo human tumor-free lung model: a comparison with the standard cardenolide is of interest and should be considered.
- Discussion: the true interest in studying novel semisynthetic cardenolides? Are there some advantages with respect to the standard cardenolides? Can the Author make some hypothesis about the structure activity relationship of C10 and C11?
Author Response
Answers to the Reviewer 2 - "Please see the attachment"
_________________________________________________________
Summary: The present study is focused on the investigation of the antiviral properties of novel semisyntetic cardenolides against different influenza strains. Among the tested compounds, C10 and C11 were highlighted to be the most promising compounds, exhibiting an anti-influenza power similar to that of the standard cardenolide digitoxigenin. The study is of interest in the field and is well designed and developed. Particularly, details about the mechanisms involved in the inhibition of viral replication have been demonstrated. The following points should be checked and clarified in order to improve the manuscript for publication.
COMMENT 1: Introduction (page 3, lines 108-114): results should be not included in this section; conversely, the aim of the study and the impact of the expected results should be reported.
Response 1: We rearranged the section (Introduction) and improved it as follows (in yellow the text added now):
Page 3, lines 107-113: “In the present study, the anti-influenza virus effects of the same 16 new cardenolide derivatives were evaluated. Since two of them showed the most significant inhibitory potential on viral replication, namely C10 {3β-[(N-(2-hydroxyethyl)aminoacetyl]amino-3-deoxydigitoxigenin} and C11 {3β-(hydroxyacetyl)amino-3-deoxydigitoxigenin}, the main goal of this work was to explore their antiviral mechanism by using different in vitro and ex vivo methodological strategies. In this sense, C10 and C11 could be considered as promising candidates to be further evaluated for the treatment of influenza virus infections”.
COMMENT 2: Materials and methods (page 3, line 120): specify that A549 is an adenocarcinoma cell line.
Response 2: The suggestion was implemented as follows (in yellow the word added now):
Page 3, line 119: “Adenocarcinomic human alveolar basal epithelial cells (A549)”
COMMENT 3: Materials and methods (page 4, line 140): the choice of the tested concentration for the antiviral activity should be explained.
Response 3: For several years, our research group has been studying the cytotoxic (on numerous human cancer cell lines) and antiviral (against different DNA and RNA viruses) actions of several cardenolides (Bertol et al., 2011; Schneider et al., 2016, 2017a, 2017b, 2018a, 2018b, and 2020; Silva et al., 2018; Boff et al., 2019, 2020a and 2020b). In this context, our experience showed that in the preliminary screenings, the ranges of action of cardenolides were similar in relation to the tested concentrations. Attesting this, the preliminary cytotoxic and antiviral screenings (Boff et al, 2019) of the same compounds tested herein disclosed an action range less than 1 µM (most active ones). For this reason, such concentration was chosen in the present study to perform the anti-influenza virus screening.
Bertol JW, Rigotto C, Pádua RM, Kreis W, Barardi CR, Braga FC, Simões CMO. Antiherpes activity of glucoevatromonoside, a cardenolide isolated from a Brazilian cultivar of Digitalis lanata. Antiviral Res 2011; 92:73-80. https://doi.org/10.1016/j.antiviral.2011.06.015.
Schneider NFZ, Geller FC, Persich L, Marostica LL, Pádua RM, Kreis W, et al. Inhibition of cell proliferation, invasion and migration by the cardenolides digitoxigenin monodigitoxoside and convallatoxin in human lung cancer cell line. Nat Prod Res 2016; 30:1327-1331. http://dx.doi.org/10. 1080/14786419.2015.1055265.
Schneider NFZ, Cerella C, Simões CMO, Diederich M. Anticancer and immunogenic properties of cardiac glycosides. Molecules 2017a; 22:E1932. http://dx.doi.org/10.3390/molecules22111932.
Schneider NFZ, Silva IT, Persich L, Carvalho A, Rocha SC, Marostica L, et al. Cytotoxic effects of the cardenolide convallatoxin and its Na,K-ATPase regulation. Mol Cell Biochem 2017b; 428:23-39. http://dx.doi.org/10.1007/s11010-016-2914-8.
Schneider NFZ, Persich L, Rocha SC, Ramos ACP, Cortes VF, Silva IT, et al. Cytotoxic and cytostatic effects of digitoxigenin monodigitoxoside (DGX) in human lung cancer cells and its link to Na,K-ATPase. Biomed Pharmacother 2018a; 97:684-696. http://dx.doi.org/10.1016/j.biopha.2017.10.128.
Schneider NFZ, Cerella C, Lee JY, Mazumder A, Kim KR, Carvalho A, et al. Cardiac glycoside glucoevatromonoside induces cancer type-specific cell death. Front Pharmacol 2018b; 9:70. http://dx.doi.org/10.3389/fphar.2018.00070.
Schneider NFZ, Menegaz D, Dagostin ALA, Persich L, Rocha SC, Ramos ACP, et al. Cytotoxicity of glucoevatromonoside alone and in combination with selected chemotherapy drugs and its effects on Na+,K+-ATPase and ion channels in lung cancer cells. Biochem Pharmacol 2020 (submitted).
Silva IT, Munkert J, Nolte E, Schneider NFZ, Rocha SC, Ramos ACP, et al. Cytotoxicity of AMANTADIG - a semisynthetic digitoxigenin derivative - alone and in combination with docetaxel in human hormone-refractory prostate cancer cells and its effect on Na+/K+-ATPase inhibition. Biomed Pharmacother 2018; 107:464-474. http://dx.doi.org/10.1016/j.biopha.2018.08.028.
Boff L, Munkert J, Ottoni FM, Schneider NFZ, Ramos GS, Kreis W, et al. Potential anti-herpes and cytotoxic action of novel semisynthetic digitoxigenin-derivatives. Eur J Med Chem 2019; 167:546-561. http://dx.doi.org/10.1016/j.ejmech.2019.01.076.
Boff, L., Schneider, N.F.Z., Munkert, J., Ottoni, F.M., Ramos, G.S., Kreis, W., Braga, F.C., Alves, R.J., Pádua, R.M., Simões, C.M.O. Elucidation of the mechanism of anti-herpes action of two novel semisynthetic cardenolide derivatives. Arch Virol. 2020a, 165, 1385-1396, https://doi.org/10.1007/s00705-020-04562-1.
Boff, L., Persich, L., Brambila, P., Ottoni, F.M., Munkert, J., Ramos, G.S., Viana, A.R.S., Kreis, W., Braga, F.C., Alves, R.J., Pádua, R.M., Schneider, N.F.Z., Simões, C.M.O. Investigation of the cytotoxic activity of two novel digitoxigenin analogues on H460 lung cancer cells. Anti-Cancer Drugs, 2020b, 31, 452-462, https://doi.org/10.1097/CAD.0000000000000872.
COMMENT 4: Materials and methods (page 4, line 150) “Only for the compounds that reduced > 50% of viral titers (digitoxigenin, C7c, C10, C11 and C12), the IC50 values were calculated”: IC50 calculation requires that a concentration-response curve be made. Please, explain the method used for the calculation of this parameter.
Response 4: For the determination of IC50 values, nonlinear regression of concentration-response curves was used (GraphPad Prism 7.00 Software, La Jolla, CA, EUA). This explanation was also added in the manuscript as follows (in yellow the text added now):
Page 4, lines 148-150: “…the IC50 values were calculated (concentration of each compound that reduced viral replication by 50%) by nonlinear regression of concentration-response curves”.
COMMENT 5: Materials and methods (page 4, line 163): change “Western” as western.
COMMENT 6: Materials and methods (page 5, line 214): specify the source of lung tissue explants and provide the ethical statement.
Response 6: We rearranged the section (2.4.4.) and improved it as follows (in yellow the text added now). We did not include the full ethics approval statement of the Deutsche Ärztekammer, because its written in German and may thus not be helpful for readers or reviewers. The approval is clearly identifiable by the code AZ: 2016-265-f-S, which is mentioned here as commonly done in publications.
Page 5, lines 212-224: “This assay was conducted as described by Matos et al. [62] and the ethical approval was given by the Ethical Council of the Deutsche Ärztekammer (AZ: 2016-265-f-S). Firstly, tumor-free human lung tissue explants were obtained from patients undergoing lung surgery at the University Hospital Muenster, Germany. All patients have given their written consent. Pieces of lung tissue were recovered in Roswell Park Memorial Institute (RPMI-1640) medium (Sigma, Germany) on the day of surgery, and stored at 4°C. Immediately after, tissues were cut in pieces of approximately 100 mg, transferred to 12-well plates containing medium, and incubated overnight at 37°C with 5% CO2. Then, tissue fragments were infected with recombinant IAV [A/Panama/2007/1999 (H3N2)] in the presence of RPMI-1640 medium [supplemented with 2 mM L-Glutamine (Sigma, Germany), 1% P/S and 0.1% BSA]. After 60 min, tissues were transferred to new 12-well-plates with medium containing C10 and C11, at 0.5, 1.0 and 2.0 µM, and were incubated for 48 h. Supernatants were collected at 1, 24 and 48 h post infection (p.i.), and stored at -80ºC for viral titer determination using plaque standard assay on MDCK cells [53,54]”.
COMMENT 7: Materials and methods (page 5, line 223): specify the method applied for IC50 calculation and software used. Moreover, the Authors should explain if data from different biological replicates are pooled for the statistical analysis and how many replicates have been made.
Response 7: The suggestion was implemented as follows (in yellow the text added now):
Page 5, lines 226-229: “The mean values ± standard deviations (SD) are representative of three or more independent experiments. For the determination of IC50 values, nonlinear regression of concentration-response curves was used. Statistical analyses were performed by ANOVA followed by post-hoc tests as indicated. All analyses were performed by using GraphPad Prism 7.00 Software, La Jolla, CA, EUA”.
COMMENT 8: Figure 1 (page 6): the statistical significance with respect the control should be included; moreover, that of C7c, C10, C11 and C12 with respect to the standard cardenolide digitoxigenin, should be included in Figure 1B. Accordingly, the statistical significance should be included in Figure S1.
Response 8: This suggestion was implemented in Figures 1A, 1B and S1.
COMMENT 9: Digitoxigenin is usually effective like C10 against all the tested strains. Are there some advantages in using the synthetic derivative C10 with respect digitoxigenin? Please, discuss.
Response 9: We have already discussed it in the Discussion section as follows:
Page 12, second paragraph, 8 to 12th lines: “As described by Boff et al. [50], digitoxigenin was used as scaffold for the semisynthesis of the derivatives cited above, and it was the most cytotoxic compound on A549 and MDCK cells, while presented similar IC50 values to those of C10 and C11. These results showed digitoxigenin as the tested compound with the lowest SI value, whilst C10 and C11 disclosed the highest SI values”.
COMMENT 10: 2. Cardenolide derivatives effects on cell viability and influenza replication (page 6): the CC50, IC50 and selectivity indices (SI) parameters should be described in the paragraph.
Response 10: We rearranged the section (3.2) and improved it as follows (in yellow the text added now):
Page 7, 1st to 6th lines: “Since digitoxigenin and compounds C7c, C10, C11 and C12 were shown to be the most active in the preliminary screenings, their cytotoxicity was evaluated on A549 and MDCK cells. None of them reduced cell viability at concentrations at least five times higher than the concentrations used in the initial screenings (Table 1). This evaluation ensured that the compounds did not act indirectly by simply reducing cell viability but rather acting directly on virus propagation by interfering with certain stages of their replication cycle. For all compounds, IC50 and CC50 values as well as the selectivity indices (SI) against IAV [strain A/WSN/33 (H1N1)] were calculated (Table 1). Compounds C10 and C11 were the most potent against IAV replication presenting the lowest IC50 values of approximately 0.06 µM [A549 cells: 0.057 µM (C10) and 0.062 µM (C11); MDCK cells: 0.060 µM (C10) and 0.066 µM (C11)], and the highest CC50 values [A549 cells: 12.790 µM (C10) and 10.025 µM (C11); MDCK cells: 18.343 µM (C10) and 15.983 µM (C11)]. In this sense, they showed SI values of 226 and 161 on A549 cells; and 306 and 241 on MDCK cells, respectively. Based on these favorable results, compounds C10 and C11, whose chemical structures are depicted in Figure 2, were selected to further explore their antiviral mechanisms of action”.
COMMENT 11: Table 1 (page 7): the IC50 differences should be calculated. Are there significant differences between the IC50 of C10, C11 and digitoxigenin?
Response 11: There are no significant differences between the IC50 values of C10, C11 and digitoxigenin on both cell lines (one-way ANOVA, post-hoc test Dunnet).
COMMENT 12: Figure 4 (page 9): time exposure for studying the effect of the test compound on influenza A virus RNA transcription is lower than that applied for the viral replication (24 h). Please, explain.
Response 12: We choose only one time point analysis (6h post infection) for the influenza A virus RNA transcription just to confirm the results presented in Figure 3 (western blott analysis), given the detected viral proteins reduction. In addition, this specific time point resembles the time of an influenza virus replication cycle (Dou et al., 2018; WHO, 2020).
Dou, D., Revol, R., Östbye, H., Wang, H., Daniels, R. Influenza A virus cell entry, replication, virion assembly and movement. Front Immunol. 2018, 9, 1581. https://doi.org/10.3389/fimmu.2018.01581.
World Health Organization (WHO). Virology of human influenza. https://www.euro.who.int/en/health-topics/communicable-diseases/influenza/data-and-statistics/virology-of-human-influenza#:~:text=The%20replication%20cycle%20of%20influenza,occurring%20after%20only%206%20hours. 2020, (accessed in August 2020).
COMMENT 13: 3.2. C10 and C11 affect the polymerase complex (page 9): a comparison with digitoxigenin should be included.
Response 13: We do not feel that a comparison with digitoxigenin would represent a major advance in this data and would even take away the attention of the reader from the fact we want to point out.
COMMENT 14: Figures 3 and 5: densitometric analysis should be included.
Response 14: This suggestion was implemented in Figures 3 and 5.
COMMENT 15: ex vivo human tumor-free lung model: a comparison with the standard cardenolide is of interest and should be considered.
Response 15: Despite the ex vivo tumor-free human lung tissue explant model used in this work is considered robust and viable economically, one of the limitations of this study was the number of lung pieces provided for the analyses, since such explants were supplied by patients who need lung surgery, and they were not numerous at the time of the study. In this sense, the preference was given to test only the focus compounds (C10 and C11), at three different concentrations and their respective controls, in order to obtain more robust results.
COMMENT 16: Discussion: the true interest in studying novel semisynthetic cardenolides? Are there some advantages with respect to the standard cardenolides? Can the Author make some hypothesis about the structure activity relationship of C10 and C11?
Response 16: The first published paper regarding the same compounds studied herein described in detail the semisynthesis of these compounds as well as the preliminary evaluation of their anti-herpes and cytotoxic activities (Boff et al., 2019). In this previous paper, we have already discussed their structure activity relationship. In addition, the main innovation of this previous paper was lied to the structural differences between the new digitoxigenin derivatives (the same tested herein) and the cardenolides tested before by our research group, which are the standard ones (Bertol et al., 2011). The cardenolides tested by Bertol et al. (2011) presented one or more sugar moieties bind to C3, differently of these new derivatives C10 and C11 that present substituted amine groups at C3. They were able to inhibit HSV (Boff et al., 2020) and influenza virus (this manuscript under submission) replication at nM concentrations, but not at the same potency of those cardenolides bearing sugars at C3. In this way, the sugar moiety seems not to be crucial to the antiviral activity, but it indeed influences this action positively. We have explained these facts in the Discussion section of Boff et al. (2020).
Boff L, Munkert J, Ottoni FM, Schneider NFZ, Ramos GS, Kreis W, et al. Potential anti-herpes and cytotoxic action of novel semisynthetic digitoxigenin-derivatives. Eur J Med Chem 2019; 167:546-561. http://dx.doi.org/10.1016/j.ejmech.2019.01.076.
Bertol JW, Rigotto C, Pádua RM, Kreis W, Barardi CR, Braga FC, Simões CMO. Antiherpes activity of glucoevatromonoside, a cardenolide isolated from a Brazilian cultivar of Digitalis lanata. Antiviral Res 2011; 92:73-80. https://doi.org/10.1016/j.antiviral.2011.06.015.
Boff, L., Schneider, N.F.Z., Munkert, J., Ottoni, F.M., Ramos, G.S., Kreis, W., Braga, F.C., Alves, R.J., Pádua, R.M., Simões, C.M.O. Elucidation of the mechanism of anti-herpes action of two novel semisynthetic cardenolide derivatives. Arch Virol. 2020, 165, 1385-1396, https://doi.org/10.1007/s00705-020-04562-1.

Round 2
Reviewer 1 Report
I must thank Boff et al. for making their submission much clearer and clarifying a number of concerns I had in regards to the original manuscript.
With the revised manuscript my main concern is that the CC50 was performed in such a small concentration range (0.078-5.0 uM) and the CC50’s “estimated” (in the range of 9-23 uM). How was this performed? Nonlinear regression? There should be more details provided although I still believe this is not an acceptable method to determine the CC50 and as such the CC50 should be reported as >5 µM.
I would also like to see the concentration curves which were used to determine the IC50’s and more information on the concentrations of compound tested.
Author Response
Answers to the Reviewer 1 - Please see the attachment
__________________________________________________________
Summary: I must thank Boff et al. for making their submission much clearer and clarifying a number of concerns I had in regard to the original manuscript.
We would like to thank the Reviewer for the careful appraisal of the manuscript. The constructive comments helped us to improve and clarify some important aspects in the revised version of the manuscript.
COMMENT 1: With the revised manuscript my main concern is that the CC50 was performed in such a small concentration range (0.078-5.0 µM) and the CC50’s “estimated” (in the range of 9-23 µM). How was this performed? Nonlinear regression? There should be more details provided although I still believe this is not an acceptable method to determine the CC50 and as such the CC50 should be reported as >5 µM.
Response 1: For several years, our research group has been studying the cytotoxic (on numerous human cancer cell lines) and antiviral (against different DNA and RNA viruses) actions of several cardenolides (Bertol et al., 2011; Schneider et al., 2016, 2017a, 2017b, 2018a, 2018b, and 2020; Silva et al., 2018; Boff et al., 2019, 2020a and 2020b). In this context, our experience showed that the CC50 ranges are typically low.
As it was explained before, due to the limitation regarding the available amount of the compounds, an estimation of the CC50 values, by nonlinear regression, was done since none of them showed cytotoxicity at the highest concentration tested (5 µM). Such estimation was made based on the eight tested concentrations (5.0, 2.5, 1.25, 0.625, 0.313, 0.156 and 0.078 µM) of digitoxigenin, C7c, C10, C11 and C12.
We chose to estimate the value instead of simply report >5 µM since one of the main objectives of this manuscript was to show one of the advantages of synthesizing new compounds less cytotoxic than its scaffold. Additionally, in a previous paper, we had already showed that digitoxigenin, used as the scaffold for the semisynthesis of the derivatives tested herein, was the most cytotoxic compound (Boff et al., 2019).
Bertol JW, Rigotto C, Pádua RM, Kreis W, Barardi CR, Braga FC, Simões CMO. Antiherpes activity of glucoevatromonoside, a cardenolide isolated from a Brazilian cultivar of Digitalis lanata. Antiviral Res 2011; 92:73-80. https://doi.org/10.1016/j.antiviral.2011.06.015.
Schneider NFZ, Geller FC, Persich L, Marostica LL, Pádua RM, Kreis W, et al. Inhibition of cell proliferation, invasion and migration by the cardenolides digitoxigenin monodigitoxoside and convallatoxin in human lung cancer cell line. Nat Prod Res 2016; 30:1327-1331. http://dx.doi.org/10. 1080/14786419.2015.1055265.
Schneider NFZ, Cerella C, Simões CMO, Diederich M. Anticancer and immunogenic properties of cardiac glycosides. Molecules 2017a; 22:E1932. http://dx.doi.org/10.3390/molecules22111932.
Schneider NFZ, Silva IT, Persich L, Carvalho A, Rocha SC, Marostica L, et al. Cytotoxic effects of the cardenolide convallatoxin and its Na,K-ATPase regulation. Mol Cell Biochem 2017b; 428:23-39. http://dx.doi.org/10.1007/s11010-016-2914-8.
Schneider NFZ, Persich L, Rocha SC, Ramos ACP, Cortes VF, Silva IT, et al. Cytotoxic and cytostatic effects of digitoxigenin monodigitoxoside (DGX) in human lung cancer cells and its link to Na,K-ATPase. Biomed Pharmacother 2018a; 97:684-696. http://dx.doi.org/10.1016/j.biopha.2017.10.128.
Schneider NFZ, Cerella C, Lee JY, Mazumder A, Kim KR, Carvalho A, et al. Cardiac glycoside glucoevatromonoside induces cancer type-specific cell death. Front Pharmacol 2018b; 9:70. http://dx.doi.org/10.3389/fphar.2018.00070.
Schneider NFZ, Menegaz D, Dagostin ALA, Persich L, Rocha SC, Ramos ACP, et al. Cytotoxicity of glucoevatromonoside alone and in combination with selected chemotherapy drugs and its effects on Na+,K+-ATPase and ion channels in lung cancer cells. Biochem Pharmacol 2020 (submitted).
Silva IT, Munkert J, Nolte E, Schneider NFZ, Rocha SC, Ramos ACP, et al. Cytotoxicity of AMANTADIG - a semisynthetic digitoxigenin derivative - alone and in combination with docetaxel in human hormone-refractory prostate cancer cells and its effect on Na+/K+-ATPase inhibition. Biomed Pharmacother 2018; 107:464-474. http://dx.doi.org/10.1016/j.biopha.2018.08.028.
Boff L, Munkert J, Ottoni FM, Schneider NFZ, Ramos GS, Kreis W, et al. Potential anti-herpes and cytotoxic action of novel semisynthetic digitoxigenin-derivatives. Eur J Med Chem 2019; 167:546-561. http://dx.doi.org/10.1016/j.ejmech.2019.01.076.
Boff, L., Schneider, N.F.Z., Munkert, J., Ottoni, F.M., Ramos, G.S., Kreis, W., Braga, F.C., Alves, R.J., Pádua, R.M., Simões, C.M.O. Elucidation of the mechanism of anti-herpes action of two novel semisynthetic cardenolide derivatives. Arch Virol. 2020a, 165, 1385-1396, https://doi.org/10.1007/s00705-020-04562-1.
Boff, L., Persich, L., Brambila, P., Ottoni, F.M., Munkert, J., Ramos, G.S., Viana, A.R.S., Kreis, W., Braga, F.C., Alves, R.J., Pádua, R.M., Schneider, N.F.Z., Simões, C.M.O. Investigation of the cytotoxic activity of two novel digitoxigenin analogues on H460 lung cancer cells. Anti-Cancer Drugs, 2020b, 31, 452-462, https://doi.org/10.1097/CAD.0000000000000872.
COMMENT 2: I would also like to see the concentration curves, which were used to determine the IC50’s and more information on the concentrations of compound tested.
Response 2: The IC50 values were calculated (concentration of each compound that reduced viral replication by 50%) by nonlinear regression of concentration-response curves using GraphPad Prism 7.00 Software, La Jolla, CA, EUA. For the calculation, we used ten tested concentrations (5.0, 2.5, 1.25, 0.625, 0.313, 0.156, 0.078, 0.039, 0.019 and 0.009 µM) of digitoxigenin, C7c, C10, C11 and C12.
Concentration curves on A549 cell line:
Concentration curves on MDCK cell line:

Reviewer 2 Report
The Authors included some improvements in the manuscript. However, I believe that comment 3 should be also included in the text. Moreover, quality and figure clarity should be improved, according to the Journal style.
Author Response
Answers to the Reviewer 2 - Please see the attachment
______________________________________________________________________
We would like to thank the Reviewer for the careful appraisal of the manuscript. The constructive comments helped us to improve and clarify some important aspects in the revised version of the manuscript.
COMMENT 1: The Authors included some improvements in the manuscript. However, I believe that comment 3 should be also included in the text. Moreover, quality and figure clarity should be improved, according to the Journal style.
Response 1: The suggestion was implemented in the Discussion section as follows (in yellow the word added now):
Page 12, second paragraph, 5 to 11th lines: “For several years, our research group has been studying the cytotoxic (on numerous human cancer cell lines) and antiviral (against different DNA and RNA viruses) actions of several cardenolides [38,41,50,51,64-69]. In this context, our experience showed that in the preliminary screenings, the ranges of action of cardenolides were similar in relation to the tested concentrations. Attesting this, the preliminary cytotoxic and antiviral screenings [50] of the same compounds tested herein disclosed an action range less than 1 µM (most active ones). For this reason, such concentration was chosen in the present study to perform the anti-influenza virus screening”.
Regarding the quality and figure clarity, the journal production office prepared the manuscript to send to the reviewers. We have no control over the production of the manuscript. But to help, we will attach here the figures sent to Biomolecules journal.
